# Diverse Effects of Cilostazol on Proprotein Convertase Subtilisin/Kexin Type 9 between Obesity and Non-Obesity

**DOI:** 10.3390/ijms23179768

**Published:** 2022-08-29

**Authors:** Po-Wei Chen, Shih-Ya Tseng, Hsien-Yuan Chang, Cheng-Han Lee, Ting-Hsing Chao

**Affiliations:** 1Division of Cardiology, Department of Internal Medicine, National Cheng Kung University Hospital, College of Medicine, National Cheng Kung University, Tainan 704, Taiwan; 2Health Management Center, National Cheng Kung University Hospital, Tainan 704, Taiwan

**Keywords:** proprotein convertase subtilisin/kexin type 9, low-density lipoprotein receptor, peroxisome proliferator-activated receptor-γ, cilostazol, obesity

## Abstract

Proprotein convertase subtilisin/kexin type 9 (PCSK9) plays a key role in cholesterol homeostasis. Cilostazol exerts favorable cellular and metabolic effects; however, the effect of cilostazol on the expression of PCSK9 has not been previously reported. Our study aimed to investigate the potential mechanisms of action of cilostazol on the expression of PCSK9 and lipid homeostasis. We evaluated the effects of cilostazol on the expression of PCSK9 in HepG2 cells and evaluated potential molecular mechanisms by measuring signaling molecules in the liver and serum lipid profiles in high-fat diet-induced obese mice and normal chow-fed mice. Cilostazol treatment significantly induced the messenger RNA and protein expression of PCSK9 in HepG2 cells and enhanced PCSK9 promoter activity. Chromatin immunoprecipitation assays confirmed that cilostazol treatment enhanced PCSK9 transcription by binding to peroxisome proliferator-activated receptor-γ (PPARγ) via the PPARγ DNA response element. PPARγ knockdown attenuated the stimulatory effect of cilostazol on PCSK9. In vitro, cilostazol treatment increased PCSK9 expression in vehicle-treated HepG2 cells but decreased PCSK9 expression in palmitic acid-treated HepG2 cells. In vivo, cilostazol treatment increased the serum levels of PCSK9 in normal mice but significantly reduced PCSK9 levels in obese mice. The expressions of PCSK9-relevant microRNAs also showed similar results. Clinical data showed that cilostazol treatment significantly reduced serum PCSK9 levels in patients with obesity. The obesity-dependent effects of cilostazol on PCSK9 expression observed from bench to bedside demonstrates the therapeutic potential of cilostazol in clinical settings.

## 1. Introduction

Proprotein convertase subtilisin/kexin type 9 (PCSK9) affects plasma lipoprotein levels and plays a crucial role in cholesterol homeostasis by increasing the degradation of hepatic low-density lipoprotein receptor (LDLR), resulting in elevated plasma levels of total cholesterol (TC) and low-density lipoprotein cholesterol (LDL-C) [1,2]. From a genetic perspective, lower plasma LDL-C levels have been observed in individuals with specific sequence variations in the PCSK9 gene, and these individuals exhibit a lower incidence of coronary artery disease (CAD) [3]. Atherogenic lipoprotein levels and many relevant cardiovascular risk factors, such as fasting plasma glucose and blood pressure, correlate with plasma PCSK9 levels [4,5]. In addition, PCSK9 has been reported to be associated with non-traditional risk factors involving oxidative and inflammatory processes [6,7,8,9,10]. Moreover, plasma PCSK9 levels are associated with CAD severity [11] and the development of severe peripheral artery disease (PAD) [12]. In our previous research, we found that circulating endothelial progenitor cell dysfunction, the number of apoptotic circulating endothelial cells (ECs), and vasculoangiogenic and oxidative biomarkers were significantly correlated with PCSK9 levels in patients with PAD [12]. Taken together, these findings indicated that the PCSK9 protein is an important target for the prevention and treatment of atherosclerosis.

Studies have shown that plasma PCSK9 levels are affected by statins [13]. The potential mechanisms underlying the role of statins have received much attention in recent years. Statins significantly increase PCSK9 concentration and hepatic LDLR expression, likely through the upregulation of sterol regulatory element-binding protein-2 (SREBP-2) [13]. Some studies have indicated a relationship between PCSK9 and the metabolism of high-density lipoprotein cholesterol (HDL-C) [14] and triglycerides (TG) [15]. Expression of PCSK9 is regulated by SREBP-2 and peroxisome proliferator-activated receptor-γ (PPARγ) [16,17]. Furthermore, the activation of the adiponectin receptor (adipoR) can regulate PCSK9 expression through the activation of PPARγ and the adenosine monophosphate-activated protein kinase (AMPK) signaling pathway [17]. 

Cilostazol, a phosphodiesterase 3 inhibitor, has been approved for the treatment of patients with PAD and intermittent claudication owing to its antiplatelet and vasodilatory effects [18,19,20]. Recently, we and other researchers have found that this compound has beneficial effects on metabolic parameters, angiogenesis, and the number and function of circulating human early endothelial progenitor cells in vitro [18,21,22,23,24], in vivo [18,21,22,23,24], and in clinical settings [19,20]. In patients with PAD or those at high risk of cardiovascular disease, cilostazol treatment can also exert significant effects on TG/HDL-C levels and may affect LDL-C levels [19,20]. However, to the best of our knowledge, no study has evaluated the effects of cilostazol on hepatic LDLR and PCSK9 or its underlying molecular mechanisms.

As a traditional risk factor for CAD, obesity also plays an important role in the development and progression of PAD. Modification of obesity-related cardiovascular risk factors is essential for reducing the risk of atherosclerotic disease [25]. Obese subjects have higher PCSK9 levels than lean controls, suggesting that PCSK9-targeting therapy may be important in such metabolic disorders [26]. Adiponectin is the most abundant hormone derived from adipose tissue. Previous studies have shown that adiponectin is a therapeutic target for obesity as higher serum concentrations of adiponectin protect against inflammation and are associated with a lower cardiovascular risk in obese subjects [27]. 

MicroRNAs (miRNAs) are a class of small, endogenous, noncoding, single-stranded RNAs [28]. The mature miRNAs negatively regulate gene expression by targeting specific mRNAs for cleavage or translational repression [28]. Detecting the levels of tissue miRNAs, as well as the levels of circulating miRNAs, is considered to be a potential approach to identifying novel biomarkers or therapeutic targets [28]. Recently, some miRNAs have been reported to be involved in the lipid metabolism [29,30,31]. However, there is neither clinical nor bench study to evaluate the modulating effect of cilostazol on lipid metabolism-relevant miRNAs.

Our previous clinical trial revealed that cilostazol treatment increases adiponectin concentrations in patients with PAD [12]. We and others have found that cilostazol can serve as an activator of AMPK signaling molecules [18,21,22,23,24], adipoRs, and PPARγ [32]. In this study, we hypothesized that cilostazol can be associated with the expression of PCSK9 and lipid metabolism through AMPKα/PPARγ signaling pathway, the effects of cilostazol treatment on PCSK9 will be different between obesity and non-obesity, and lipid metabolism-relevant miRNAs will partly contribute to the diverse effects.

## 2. Results

### 2.1. Cilostazol Induces PPARγ Upregulation and PCSK9 Expression in HepG2 Cells

To determine the effect of cilostazol on PPARγ and PCSK9 expression, HepG2 cells were treated with varying concentrations of cilostazol in serum-free medium overnight. The expression levels of PPARγ and PCSK9 proteins were assessed using Western blotting. As shown in Figure 1A, cilostazol treatment induced PPARγ and PCSK9 expression in HepG2 cells in a concentration-dependent manner. The maximal levels of induction were observed between 16 and 24 h post-treatment (Figure 1B). Similarly, cilostazol treatment induced PPARγ and PCSK9 transcription in a dose-dependent manner in HepG2 cells (Figure 1C,D). Furthermore, cilostazol enhanced PPARγ promoter activity (Figure 1E). Both the peroxisome proliferator response element (PPRE) and sterol regulatory element (SRE) are present in the human PCSK9 promoter region. To determine the role of cilostazol in PCSK9 transcription, we constructed PCSK9 promoters with a mutation in either the PPRE (pPCSK9-PPREmut) or SRE (pPCSK9-SREmut). Cilostazol enhanced PCSK9 promoter activity in control cells and this effect was attenuated in cells transfected with the PPRE-mutated PCSK9 promoter, but not the SRE-mutated PCSK9 promoter (Figure 1F). As shown in Figure 1G, a chromatin immunoprecipitation (ChIP) assay revealed that cilostazol treatment enhanced PCSK9 transcription by binding to PPARγ via the PPRE, but not to SREBP-2 via the SRE. We transiently transfected HepG2 cells with scrambled small interfering RNA (siRNA; control) and PPARγ siRNA and demonstrated that downregulation of PPARγ via transfection with PPARγ siRNA attenuated the stimulatory effect of cilostazol on PCSK9 expression, while transfection with the scrambled siRNA did not (Figure 1H). Therefore, our data suggest that PPARγ mediates the effects of cilostazol on PCSK9 expression at the transcriptional level.

### 2.2. Cilostazol Induces LDLR Expression in HepG2 Cells

The expression of LDLR protein was enhanced following cilostazol treatment, with the greatest effect at 16–24 h post-treatment, as revealed by Western blotting and immunofluorescence assays (Figure 2A–C). Furthermore, LDLR mRNA was upregulated by cilostazol treatment (Figure 2D). To explore how cilostazol regulates LDLR expression, we constructed an LDLR promoter with an SRE mutation (pLDLR-SREmut). This mutation diminished the effect of cilostazol treatment on LDLR promoter activity (Figure 2E). A ChIP assay revealed that the interaction between SREBP-2 and SRE was enhanced by cilostazol treatment (Figure 2F). As shown in Figure 2G, cilostazol treatment increased both the precursor and mature forms of SREBP-2. Taken together, our data indicate that cilostazol enhances SREBP-2 expression and maturation, thereby leading to the induction of LDLR expression.

### 2.3. Cilostazol Mediates the Expression of Adiponectin and the Subsequent Activation of AMPKα/PPARγ Signaling Molecules in HepG2 Cells 

To determine the effect of cilostazol on the adiponectin/adipoR/AMPK/PPARγ /PCSK9 axis, we measured mRNA and protein levels of adiponectin and found that both were increased following cilostazol in HepG2 cells (Figure 3A,B). Subsequently, we transfected HepG2 cells with lentivirus-mediated short hairpin (sh) RNA against adipoRs to knockdown the expression of adipoRs (Figure 3C). We observed that cilostazol treatment increased the expression of PCSK9 and LDLR and the phosphorylation of AMPKα; these effects were attenuated following the knockdown of the adipoRs, especially adipoR1 (Figure 3D). The knockdown of the AMPKα gene in HepG2 cells attenuated the stimulatory effects of cilostazol on the expression of PPARγ and PCSK9 (Figure 3E). From these data, we conclude that the adiponectin/adipoR/AMPK axis plays an important role in the cilostazol-induced expression of PPARγ/PCSK9.

### 2.4. Diverse Effects of Cilostazol on PCSK9 Levels in HepG2 Cells Treated with Palmitic Acid (PA) or without

By treating HepG2 cells with PA, we established an in vitro model of obesity-related fatty liver. We found that cilostazol treatment increased PCSK9 expression in vehicle-treated HepG2 cells but decreased PCSK9 expression in PA-treated HepG2 cells (Figure 4A).

In both of the vehicle-treated HepG2 cells and PA-treated HepG2 cells, SREBP-2 and pAMPKα/AMPKα were significantly increased after cilostazol treatment. PPARγ and LDLR were significantly increased in vehicle-treated HepG2 cells. In addition, cilostazol treatment was associated with numerically higher PPARγ and LDLR levels without statistical significance in PA-treated HepG2 cells.

### 2.5. Cilostazol Attenuates Lipid Accumulation in the Livers of Obese Mice

To determine the effect of cilostazol on lipid metabolism in the liver, we used high-fat diet (HFD)-fed mice to induce obesity and normal chow (NC)-fed mice as the controls. Wild-type mice were intraperitoneally injected with vehicle or 10 mg/kg cilostazol twice daily for three months. NC-fed mice had lower body weight than HFD-fed mice. A significant reduction in body weight and liver weight was observed in HFD-fed mice treated with cilostazol. In NC-fed mice, body and liver weights were similar in animals with or without cilostazol treatment (Figure 5A,B). Cilostazol treatment attenuated HFD-induced accumulation of lipids in the liver grossly (Figure 5C). Hematoxylin and eosin and Oil Red O staining of liver sections revealed abundant accumulation of lipid droplets accompanied by hepatocyte damage in HFD-fed mice. This hazardous effect was diminished by cilostazol treatment in HFD-fed mice (Figure 5D).

Compared to NC-fed mice treated with vehicle, cilostazol-treated NC-fed mice exhibited a significant increase in HDL-C levels and a decrease in TC levels (Figure 5E). Serum adiponectin and LDLR levels were not significantly different between the groups (Figure 5F,G). In addition, serum PCSK9 levels were significantly higher in cilostazol-treated NC-fed mice (Figure 5H). Western blotting demonstrated that cilostazol treatment increased the protein levels of PPARγ, LDLR, SREBP-2, upregulated the phosphorylation of AMPKα, and increased PCSK9 protein expression in the liver of the mice fed with NC (Figure 5I,J).

Compared to the NC-fed mice, cilostazol treatment produced opposite effects on lipid homeostasis in HFD-fed mice. Compared to HFD-fed mice treated with vehicle, cilostazol-treated mice exhibited a significant increase in HDL-C levels and a significant reduction in serum TC levels (Figure 6A). Serum levels of adiponectin and LDLR were significantly higher in cilostazol-treated HFD-fed mice compared to those in vehicle-treated controls (Figure 6B,C), whereas serum PCSK9 levels were significantly lower (Figure 6D). Cilostazol treatment significantly increased serum apolipoprotein A1 (ApoA1) and ApoE levels at 3 and 2 months post-treatment, respectively (Figure 6E,F). In addition, cilostazol treatment was associated with numerically lower serum ApoB levels without statistical significance (Figure 6G). The mRNA expression levels of PPARγ and LDLR were significantly higher in the cilostazol treatment group than those in the vehicle group (Figure 6H). Western blotting demonstrated that cilostazol treatment increased the protein levels of PPARγ, LDLR and SREBP-2, upregulated the phosphorylation of AMPKα, and reduced PCSK9 protein expression (Figure 6I,J). 

Taken together, these results indicate that cilostazol treatment can improve lipid metabolism in HFD-induced obese mice and regulate the expression of AMPK/PPARγ/PCSK9 and SREBP-2/LDLR signaling molecules.

### 2.6. Cilostazol Treatment Provides Anti-Atherosclerosis Effects in Mice with HFD-Induced Obesity and Shows Diverse Effects on PCSK9 Expression in Obese and Non-Obese Participants

Cilostazol treatment ameliorated atherosclerotic lesions in the aorta (Figure 7A,B). Immunofluorescence staining demonstrated that cilostazol-treated mice exhibited an increase in the expression of CD31, an EC marker, in the intima layer, whereas the expression of α-smooth muscle actin (α-SMA), a smooth muscle cell marker, was attenuated in the media layer of the aorta (Figure 7C,D). Thus, our data indicate that cilostazol may inhibit atherosclerosis. 

To examine the diverse effects of cilostazol on PCSK9 expression in obese and non-obese participants, we analyzed PCSK9 data from a prospective clinical trial and found that cilostazol treatment significantly reduced serum PCSK9 levels in patients with obesity (369.75 ± 30.94 vs. 243.27 ± 23.36 ng/mL, *p* < 0.001) but not in those without obesity (276.26 ± 24.57 vs. 243.98 ± 32.14 ng/mL, *p* = 0.178). The difference in the serum concentrations of PCSK9 between obese and non-obese patients post-treatment was statistically significant (ΔPCSK9: 126.48 ± 24.93 vs. 32.28 ± 23.68 ng/mL, *p* = 0.009) (Figure 7E).

### 2.7. Diverse Effects of Cilostazol on PCSK9-Relevant miRNAs in the Liver of the Mice Fed with NC and HFD 

To explore the potential mechanisms of action responsible for the diverse effects, we investigated the effects of cilostazol on lipid metabolism-relevant miRNAs, in the liver of the mice fed with NC and HFD.

Compared to HFD-fed mice treated with vehicle, the relative quantification of the miRNA-191 level was significantly higher in cilostazol-treated HFD-fed mice (Figure 8A). In contrast, miRNA-191 levels were not significantly different with cilostazol treatment in the NC-fed group. Moreover, miRNA-27a and miRNA-185 levels were also not significantly changed with cilostazol treatment either in HFD- or NC-fed mice (Figure 8B,C). 

Taken together, the reduction in PCSK9 expression with cilostazol treatment in obese condition might be mediated by the upregulation of some PCSK9-relevant miRNAs.

## 3. Discussion

We identified, for the first time, that cilostazol can modulate PCSK9 expression by upregulating the adiponectin/adipoR/AMPK/PPARγ signaling pathway but not by SREBP-2/SRE pathway. Moreover, cilostazol enhances LDLR expression by stimulating SREBP-2 expression and binding to the SRE of LDLR. Interestingly, diverse effects of cilostazol on PCSK9 expression were observed between obesity and non-obesity from bench (in vitro and in vivo) to bedside. 

Circulating adiponectin concentrations are reduced in obese individuals, and this reduction has been proposed to play a crucial role in the pathogenesis of atherosclerosis and cardiovascular diseases associated with obesity and lipid metabolism [33]. We and others have demonstrated that cilostazol treatment can increase plasma adiponectin levels in patients with PAD [12,34]; however, the mechanisms of action were unknown. In the present study, we demonstrated that cilostazol treatment can enhance adiponectin expression in hepatocytes and that the adiponectin/adipoR/AMPK axis plays an important role in cilostazol-induced PCSK9 expression under normal conditions.

Our data showed that cilostazol treatment had diverse effects on PCSK9 in obese and non-obese conditions. A previous study demonstrated the diverse effects of adipoR activation on PCSK9 expression in apoE-deficient (apoE^−/−^) mice and wild-type mice [17]. The current clinical data, along with the contradictory results of a previous clinical study [35], imply that different clinical settings may determine the diverse effects of cilostazol treatment on PCSK9 expression, which may partly explain the inconclusive results regarding the effects of cilostazol on TC and LDL-C levels in previous clinical studies [19,20,34,35,36]. Despite the obesity-dependent effects of cilostazol on PCSK9 expression, we observed similar effects on the expression of AMPKα and PPARγ, suggesting that mechanisms beyond AMPK/PPARγ activation may be at play. Elucidating additional mechanisms of cilostazol action is an area of active investigation by our group. 

SREBP-2 is a paradoxical regulator of plasma LDL levels. Low intracellular cholesterol levels in response to statin treatment activate SREBP-2, resulting in the co-expression of LDLR and PCSK9 [7]. Although this self-regulatory mechanism contributes to cholesterol homeostasis to prevent excessive cholesterol uptake, it may limit the therapeutic effects of statins [7]. In addition to upregulating LDLR transcription, which ultimately increases the clearance of LDL from the bloodstream, nuclear SREBP-2 increases the transcription of PCSK9, a sterol-responsive protein that accelerates LDLR turnover in the liver, thereby limiting plasma LDL-C uptake. Thus, two opposing effects on plasma cholesterol levels are initiated by the same metabolic signal. Our data demonstrated that cilostazol treatment enhanced LDLR expression by stimulating SREBP-2 and binding to the SRE of LDLR under normal conditions and was associated with higher serum LDLR levels in obese mice. However, our data showed that cilostazol treatment could modulate PCSK9 expression rather than by SREBP-2/SRE pathway. This entanglement of signaling pathways could further explain the inconclusive results regarding the effects of cilostazol on TC and LDL-C levels [19,20,34,35,36]. Nevertheless, in agreement with previous studies [17], our findings demonstrate that cilostazol has an anti-atherosclerotic effect, particularly in obese conditions. 

Previous studies have reported that cilostazol ameliorates dyslipidemia [36]. In addition, earlier studies performed in Japan and the United States demonstrated a beneficial effect of cilostazol on lipoprotein metabolism, characterized by an increase in HDL-C and a reduction in plasma triglyceride levels [36,37,38]. However, the mechanisms underlying the pharmacological efficacy of cilostazol on the complete lipid profile have not been fully elucidated in the liver [38]. Our study has revealed other mechanisms of action responsible for cilostazol-related favorable effects on the lipid profile and anti-atherosclerosis in obese conditions by reducing PCSK9 and enhancing LDLR, supporting its therapeutic application in such a clinical setting.

This is the first study to demonstrate that cilostazol can increase PCSK9 in HepG2 cells in normal conditions but decrease PCSK9 in PA-treated HepG2 cells and HFD-fed mice. The findings regarding the diverse effects of cilostazol on PCSK9 expression between obese and non-obese conditions (in vitro, in vivo, or clinically) are novel, suggesting another mechanisms of action are responsible for the anti-atherosclerotic effects of cilostazol for obesity. Our study implies that other potential mechanisms of action are responsible for cilostazol-related favorable effects on the lipid profile and PCSK9 levels in obese conditions apart from PPARγ/AMPK signaling pathway. Our results demonstrate that some PCSK9-relevant miRNA can be upregulated with cilostazol treatment in the liver of HFD-fed mice but not in NC-fed mice. Taken together, it implies that cilostazol treatment might reduce PCSK9 expression by upregulating some relevant miRNAs in obese individuals. The exact mechanism behind the phenomenon warrants further investigation.

Obesity and dyslipidemia play a pivotal role in the development and progression of CAD and PAD. Modification of obesity-related cardiovascular risk factors is essential for reducing the risk of atherosclerotic disease and PCSK9-targeting therapy may be important in such metabolic disorders. In addition to statin and PCSK9 inhibitor treatment, our data indicated another therapeutic potential, regarding cilostazol as “drug repurposing” for reducing the risk of atherosclerotic disease in obese patients.

## 4. Materials and Methods

### 4.1. Cell Culture and Preparation

The human hepatic cell line HepG2 was purchased from American Type Culture Collection (ATCC; Manassas, VA, USA) and cultured in complete Dulbecco’s modified Eagle’s medium (Gibco, Grand Island, NY, USA) supplemented with 10% fetal bovine serum, 4.5 g/L l-glutamine-sodium pyruvate, and 50 μg/mL penicillin/streptomycin (Gibco, Grand Island, NY, USA). Cells were grown in an environment with 5% CO_2_ and 95% humidity at 37 °C. Cells at 90% confluence were switched to serum-free medium before the indicated treatment.

To induce fatty liver, a common phenomenon in obesity, PA (Sigma, St. Louis, MO, USA) was dissolved in 50% ethanol by heating at 50 °C and then diluted with PBS at a 1:2 ratio. HepG2 cells were pretreated with or without 250 μM PA and incubated in the culture media for 24 h.

### 4.2. Western Blotting Assay

Western blotting was used to determine the protein expression levels of PCSK9, LDLR, SREBP-2, and PPARγ. Cells were washed twice with ice-cold phosphate-buffered saline (PBS) and solubilized in RIPA lysis buffer (Tools, Taiwan) with a protease inhibitor cocktail tablet (Roche, Branford, CT, USA). Total protein samples (30 µg/well) were electrophoresed on 8–10% sodium dodecyl sulfate-polyacrylamide gels and transferred to polyvinylidene difluoride membranes (Millipore, Darmstadt, Germany), which were blocked with 5% skimmed milk diluted with Tris-buffered saline and Tween. Membranes were incubated with primary antibodies (1:1000 dilution) against PCSK9, PPARγ (Proteintech, Chicago, IL, USA), LDLR (Cayman Chemical, Ann Arbor, MI, USA), and SREBP-2 (Novus Biologicals, CO, USA) at 4 °C overnight. The target proteins were visualized using horseradish peroxidase (HRP)-conjugated secondary antibodies (Jackson ImmunoResearch, West Grove, PA, USA) and enhanced with chemiluminescence kits (PerkinElmer, Waltham, MA, USA). Subsequently, the blots were exposed to X-ray film (Fujifilm Medical, Stamford, CT, USA). β-actin was used as an internal control.

### 4.3. Immunofluorescence Staining

LDLR expression in HepG2 cells was detected by immunofluorescence staining. HepG2 cells were seeded at a density of 5 × 10^4^ cells on a glass slide overnight. Cells treated with the indicated concentrations of cilostazol for 24 h were washed with PBS, fixed in 4% paraformaldehyde, and permeabilized with 1% bovine serum albumin and 0.2% Triton X-100. The cells were then incubated with the primary antibody against LDLR (1:200) at 4 °C overnight, followed by incubation with the appropriate Alexa Fluor^®^ 546-conjugated secondary antibodies (1:400, Thermo Fisher Scientific, Waltham, MA, USA). Cells were co-stained with 4′,6-diamidino-2-phenylindole (Sigma, St. Louis, MO, USA) to label the cell nuclei. Fluorescence was observed under a laser scanning confocal microscope (DP72; Olympus Corporation, Tokyo, Japan).

### 4.4. Quantitative Real-Time Polymerase Chain Reaction (RT-PCR)

To determine the mRNA expression levels of PCSK9, PPARγ, and LDLR, total RNA was extracted from HepG2 cells using TRIzol (Invitrogen, Carlsbad, CA, USA), according to the manufacturer’s instructions. Complementary DNA was synthesized from 2 μg total RNA using a reverse transcription kit (Thermo Fisher Scientific, Waltham, MA, USA). All samples were processed in triplicate using SYBR Green Master Mix (Qiagen, Hilden, Germany) and the ABI7500 sequence detection system (Applied Biosystems, Foster City, CA, USA). Melting curve analysis was performed to confirm the presence of specific PCR products. Gene expression levels were quantified relative to GADPH expression using the comparative Ct (2^−ΔΔCt^) value method. The primer sequences used for RT-PCR are listed in Appendix A.

For miRNA quantification, 10 ng of total RNA was used for miRNA-specific cDNA synthesis with TaqMan MicroRNA RT kit and quantitative RT-PCR was performed with the TaqMan MicroRNA Assay following the manufacturer’s protocol (Life Technologies, Carlsbad, CA, USA). Each reaction was run in triplicate. U6 was detected as the internal control.

### 4.5. siRNA Transfection

Transient transfection of HepG2 cells was performed using siRNAs. Briefly, 24 h after plating, cells were transfected with PPARγ siRNA (Santa Cruz Biotechnology, Santa Cruz, CA, USA) or an siRNA scrambled control (Santa Cruz Biotechnology, Santa Cruz, CA, USA) (50 or 100 nM) using Lipofectamine 2000 (Invitrogen, Carlsbad, CA, USA). At 48 h after transfection, the levels of PPARγ mRNA/protein were analyzed by quantitative RT-PCR and Western blotting.

### 4.6. Effects of Cilostazol on Promoter Activity with or without PPRE or SRE Mutation

Expression vectors for human PPARγ, PCSK9, and LDLR promoters were constructed using genomic DNA isolated from HepG2 cells and cloned using PCR. Next, through site-directed mutagenesis, the PCSK9 promoter was cloned with a PPRE or SRE mutation (pPCSK9-PPREmut or pPCSK9-SREmut, respectively), and the LDLR promoter was cloned with an SRE mutation (pLDLR-SREmut). The DNA was ligated into the pGL4.54 luciferase reporter vector (Promega, Madison, WI, USA). To determine promoter activity, HepG2 cells were transfected with human PPARγ, PCSK9, and LDLR promoter vectors and co-transfected with Renilla (for internal normalization) using Lipofectamine 2000 (Invitrogen, MA, USA). Transfected cells were switched to serum-free medium and treated with or without cilostazol for 24 h. The activity of promoters was determined using the Nano-Glo^®^ Dual-Luciferase Reporter Assay System (Promega, Madison, WI, USA). The sequences of the primers used for promoter constructs are listed in Appendix A.

### 4.7. ChIP Assay

DNA binding activity was evaluated using a ChIP assay kit (Millipore, Billerica, MA, USA) according to the manufacturer’s protocol. After the indicated treatment, the cells were harvested and treated with 1% formaldehyde, and nuclear proteins were extracted using lysis buffer. The cells were sonicated to produce 200–900-bp DNA fragments. Chromatin was immunoprecipitated overnight with the indicated antibody (Cell Signaling Technology) or negative control IgG. Input PCR was conducted using the DNA extracted from sonicated chromatin after the reversal of cross-linking. Immunoprecipitation was conducted using an anti-PPARγ polyclonal antibody or normal IgG, followed by PCR with primers. The sequences of the ChIP primers are listed in Appendix A.

### 4.8. Animal Model

The protocol for the animal experiments was approved by the Ethics Committee of the National Cheng Kung University, and this study conformed to the Guide for the Care and Use of Laboratory Animals published by the NIH (IACUC No. 109059). Wild-type C57BL/6 mice were purchased from the Animal Center of the National Cheng Kung University College of Medicine. All mice were maintained in a pathogen-free animal facility at 22 °C, with a relative humidity of 40–50%, in a 12 h light–dark cycle. Eight-week-old male mice were divided into NC (LabDiet (St. Louis, MO, USA), #5010, 58% kcal from carbohydrate, 29% kcal from protein, and 13% kcal from fat) or HFD (Research Diets (New Brunswick, NJ, USA), D12492, 20% kcal from carbohydrate, 20% kcal from protein, and 60% kcal from fat, total 5.24 kcal/gm) groups. The daily intake of food was the same across the groups. Mice fed the NC or HFD for 12 weeks were intraperitoneally injected with cilostazol (10 mg/kg body weight) or vehicle (2% dimethyl sulfoxide/PBS) twice a week for 12 weeks. At the end of the experiment, mice were fasted for 12 h and euthanized in a CO_2_ chamber, followed by the collection of tissue and blood samples [38].

### 4.9. Serum Biochemical Analysis

Serum samples were separated by centrifugation at 3500× *g* for 10 min at 4 °C. Serum levels of TC, TG, LDL-C, and HDL-C were examined using an automatic biochemistry analyzer (Beckman CX5; Beckman Coulter, CA, USA). Serum levels of PCSK9, LDLR, adiponectin, ApoA1, ApoB, and ApoE in experimental mice were detected using an enzyme-linked immunosorbent assay (ELISA) kit according to the manufacturer’s instructions. Absorbance was measured at a wavelength of 450 nm and plotted against standard curves.

### 4.10. Histology and Oil Red O Staining

Mouse liver tissues were fixed in 4% paraformaldehyde. A piece of liver was embedded in paraffin and cut into 5 μm-thick sections, which were stained with hematoxylin and eosin. A piece of liver was snap frozen in liquid nitrogen and the frozen sections were stained with Oil Red O (Sigma-Aldrich, St. Louis, MO, USA) and counterstained with hematoxylin. Tissue sections were imaged using a light microscope (Olympus BX-60; Olympus, Tokyo, Japan).

### 4.11. Clinical Data

We previously conducted a prospective, single-center, double-blind, double-dummy, superior, randomized, placebo-controlled trial. The study protocol (version 4.0; 21 October 2015) was approved by the IRB of the National Cheng Kung University Hospital (identifier: A-BR-102-076) and registered at ClinicalTrials.gov (identifier: NCT 02174939). This study consecutively enrolled 266 patients with stable CAD or a high risk for cardiovascular disease. Eligible patients were randomly assigned to receive cilostazol (100 mg twice daily) or a matching dummy placebo twice daily for 12 weeks. Pre-specified clinical endpoints, including the composite major adverse cardiovascular event (cardiovascular death, non-fatal myocardial infarct, non-fatal stroke, hospitalization for heart failure, or unplanned coronary revascularization), were prospectively assessed [39].

A post hoc analysis of 104 participants who had complete baseline and post-treatment serum PCSK9 data from 134 cilostazol-treated patients was performed and those participants were categorized into obesity (*n* = 43) or non-obesity (*n* = 61) groups based on the local definition of obesity (body mass index ≥ 27) in Taiwan [40]. The serum concentrations of PCSK9 were measured using commercial kits (R & D Systems Inc., Minneapolis, MN, USA).

### 4.12. Statistical Analysis

Data from at least three independent experiments were analyzed and expressed as the mean ± standard error of the mean. The differences between two groups were analyzed by unpaired Student *t*-test. The differences among three or more groups were analyzed by one-way ANOVA, followed by a Tukey’s multiple comparison test. Statistical analysis was performed using GraphPad Prism 6 (GraphPad Software Inc., San Diego, CA, USA) and the Statistical Package for the Social Sciences (SPSS) version 17.0 (SPSS Inc., Chicago, IL, USA). All values of *p* < 0.05 were considered statistically significant.

## 5. Conclusions

In conclusion, cilostazol treatment significantly modulated PCSK9 expression by upregulating adiponectin/adipoR/AMPK/PPARγ signaling molecules but not by SREBP-2/SRE pathway. Moreover, cilostazol also enhanced LDLR expression by stimulating SREBP-2 expression and binding to the SRE of LDLR. Furthermore, the diverse effects of cilostazol on PCSK9 expression were observed between obese and non-obese individuals from bench (in vitro in human hepatic cells and in vivo in mice) to bedside (clinical data). Cilostazol had a favorable effect on lipid profiles and anti-atherosclerosis in obese subjects by reducing PCSK9 and enhancing LDLR expression, supporting its therapeutic application in such clinical settings.

## Figures and Tables

**Figure 1 ijms-23-09768-f001:**
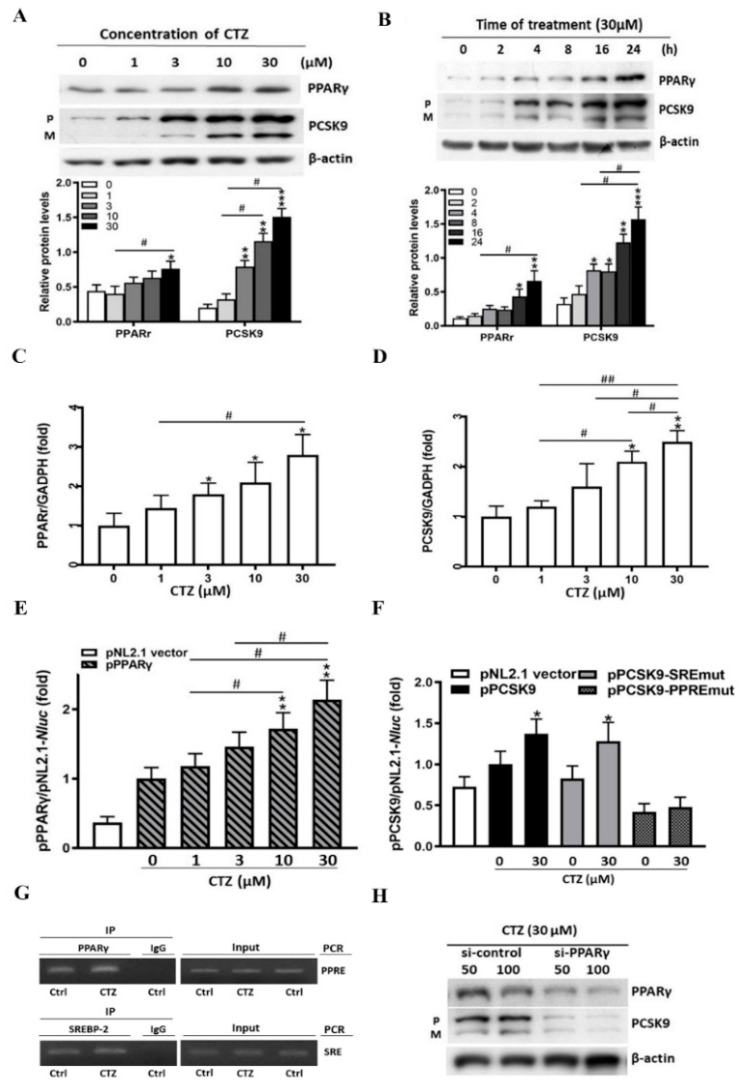
Cilostazol activates the expression of peroxisome proliferator-activated receptor-γ (PPARγ) and proprotein convertase subtilisin/kexin type 9 (PCSK9) in HepG2 cells. * *p* < 0.05, ** *p* < 0.01, and *** *p* < 0.001 versus the untreated group. # *p* < 0.05, and ## *p* < 0.01, comparison between treated groups. (**A**) HepG2 cells in serum-free medium were treated with cilostazol at different concentrations overnight (0, 1, 3, 10, 30 μM). Protein levels of PPARγ and PCSK9 in HepG2 were measured by Western blotting. β-actin was used as a loading control. Quantification of blot intensity relative to loading control is shown below. (**B**) HepG2 cells in serum-free medium were treated with cilostazol (30 μM), and then incubated for 0 min, 2 h, 4 h, 8 h, 16 h, and 24 h. Quantification of blot intensity relative to loading control is shown below. (**C**,**D**) Quantification of PPARγ and PCSK9 mRNA expression in the cilostazol-treated groups. The total cellular RNA was used to determine PPARγ and PCSK9 mRNA expressions by Cell-RNA-cDNA real-time polymerase chain reaction (PCR). The results are represented as the mean ± S.E.M. from three independent experiments. Cilostazol induced mRNA expression dose-dependently in HepG2 cells. (**E**) Quantification of PPARγ promoter activity in the cilostazol-treated groups. Expression vector of human PPARγ promoters were constructed by PCR with the genomic DNA isolated from HepG2 cells. The results are represented as the mean ± S.E.M. from three independent experiments. (**F**) Cells were transfected with DNA for PCSK9 promoter and expression vector of wild type or peroxisome proliferator response element (PPRE) or sterol regulatory element (SRE) mutation (pPCSK9-PPREmut or pPCSK9-SREmut) followed by treatment with cilostazol at the indicated concentrations overnight. (**G**) HepG2 cells were treated with cilostazol (30 μM) overnight. Chromatin was isolated followed by immunoprecipitation with normal immunoglobulin G (IgG), anti-PPARγ, or anti-sterol regulatory element-binding protein-2 (SREBP-2) antibody. The PCR was conducted with primers for PPRE or SRE in the PCSK9 promoter. (**H**) Scrambled small-interfering RNA (siRNA) and PPARγ siRNA were transfected in HepG2 cells at the indicated concentrations. The expression levels of PPARγ and PCSK9 protein were assessed by Western blot.

**Figure 2 ijms-23-09768-f002:**
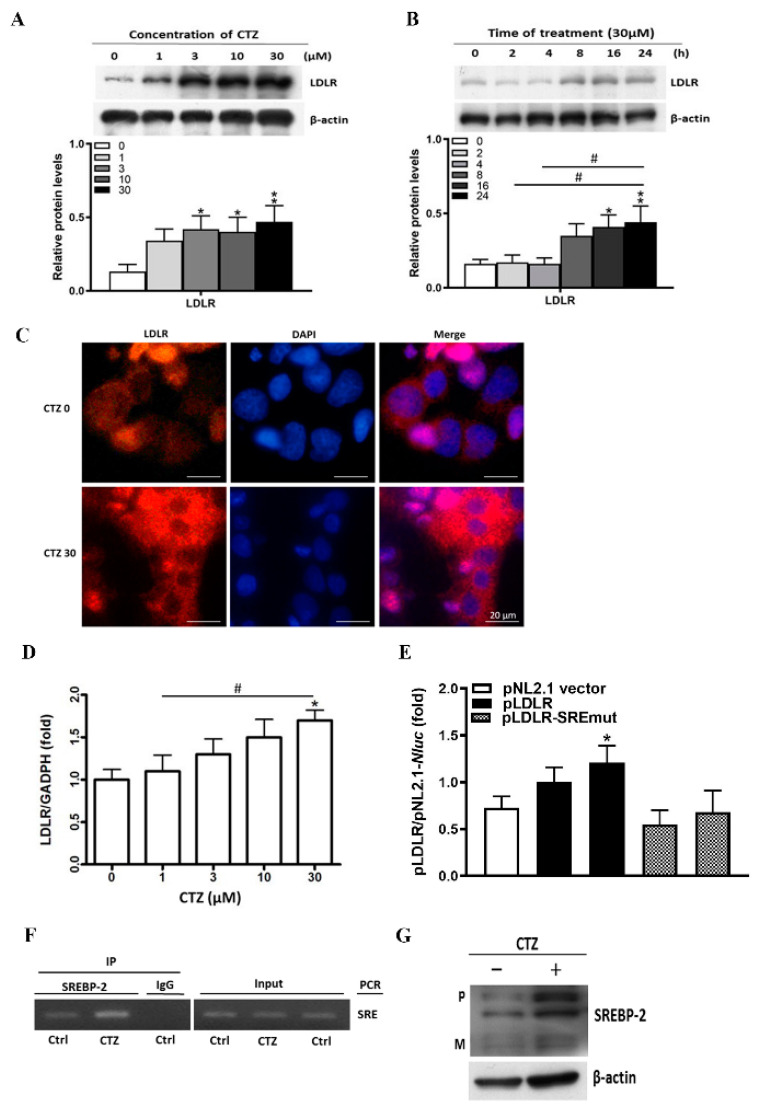
Cilostazol activates low-density lipoprotein receptor (LDLR) expression in HepG2 cells. * *p* < 0.05, and ** *p* < 0.01 versus the untreated group. # *p* < 0.05, comparison between treated groups. (**A**) HepG2 cells in serum-free medium were treated with cilostazol at different concentrations overnight (0, 1, 3, 10, 30 μM). Protein levels of LDLR in HepG2 were measured by Western blotting. β-actin was used as a loading control. Quantification of blot intensity relative to loading control is shown below. (**B**) HepG2 cells in serum-free medium were treated with cilostazol (30 μM), and then incubated for 0 min, 2 h, 4 h, 8 h, 16 h, and 24 h. Quantification of blot intensity relative to loading control is shown below. (**C**) Representative images of immunofluorescence staining with LDLR (red), 4′,6-diamidino-2-phenylindole (DAPI) (nucleus, blue), and their colocalization (merge). Scale bar = 20 μM. (**D**) Quantification of LDLR mRNA expression in the cilostazol-treated groups. The total cellular RNA was used to determine LDLR mRNA expression by Cell-RNA-cDNA real-time PCR. The results are represented as the mean ± S.E.M. from three independent experiments. (**E**) LDLR promoter was conducted with the mutation of SRE (pLDLR-SREmut). Expression vector of human LDLR promoters were constructed by PCR with the genomic DNA isolated from HepG2 cells. Cells were transfected with DNA for LDLR promoter and expression vector of wild type or SRE mutation (pLDLR-SREmut) followed by treatment with cilostazol at the indicated concentrations overnight. (**F**) HepG2 cells were treated with cilostazol (30 μM) overnight. Chromatin was isolated followed by immunoprecipitation with normal IgG or anti-SREBP-2 antibody. The PCR was conducted with primers for SRE in the PCSK9 promoter. (**G**) Protein levels of precursor (P) and mature (M) forms of SREBP-2 in HepG2 were measured by Western blotting. β-actin was used as a loading control.

**Figure 3 ijms-23-09768-f003:**
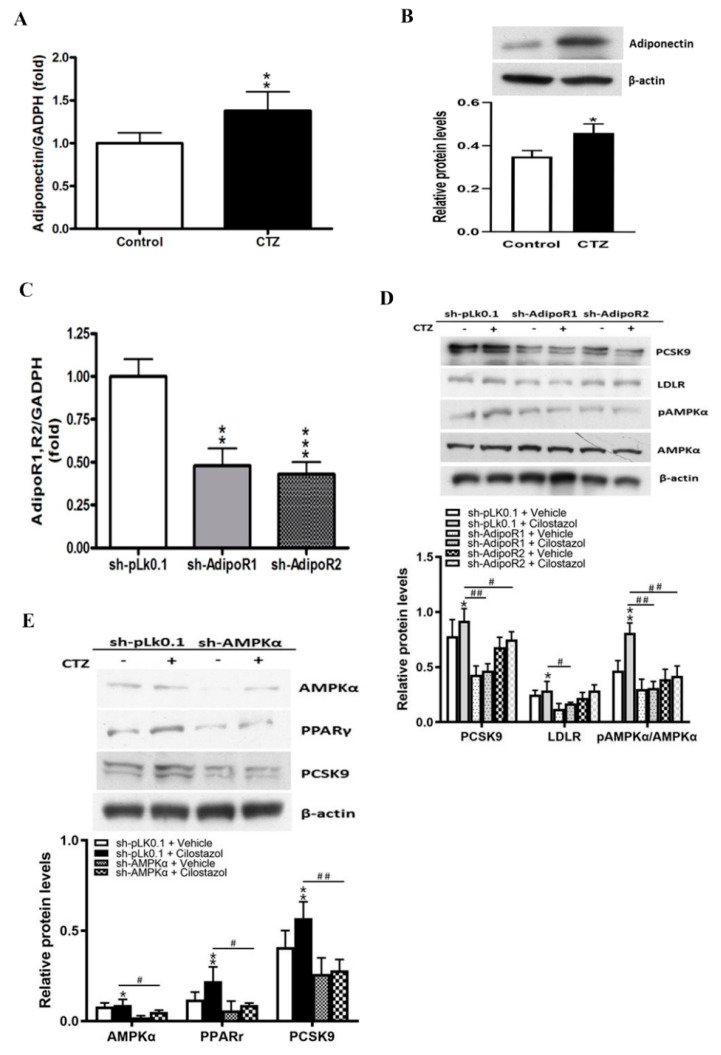
Cilostazol mediates the expression of adiponectin and the subsequent activation of adenosine monophosphate-activated protein kinase (AMPK) in HepG2 cells. * *p* < 0.05, ** *p* < 0.01, and *** *p* < 0.001 versus the control group or sh−pLK0.1+vehicle. # *p* < 0.05, ## *p* < 0.01, comparison between treated groups. (**A**) Quantification of mRNA expression of adiponectin in the cilostazol-treated groups. (**B**) Protein levels of adiponectin in HepG2 cells were measured by Western blotting. β-actin was used as a loading control. (**C**) The sh-adipoR1 and sh-adipoR2 were transfected into HepG2 cells to knock down adipoRs. (**D**) Protein expression of phosphorylated AMPKα and PCSK9 and LDLR were measured by Western blotting. Β-actin was used as a loading control. Quantification of blot intensity relative to the loading control is shown below. (**E**) Protein levels of AMPKα, PPARγ, and PCSK9 in HepG2 cells were measured by Western blotting. β-actin was used as a loading control. Quantification of blot intensity relative to loading control is shown below.

**Figure 4 ijms-23-09768-f004:**
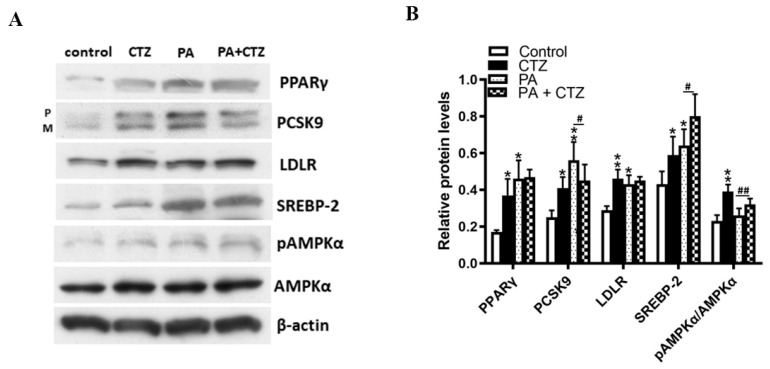
Diverse effects of cilostazol on PCSK9 levels in HepG2 cells treated with palmitic acid (PA) or without. (**A**) HepG2 cells were pretreated with or without 250 μM PA incubated in the culture media for 24 h. Protein levels of PPARγ, PCSK9, LDLR, SREBP-2, and phosphorylated AMPKα in HepG2 were measured by Western blotting. β-actin was used as a loading control. (**B**) Quantification of blot intensity relative to loading control is shown. * *p* < 0.05 and ** *p* < 0.01 versus vehicle control. # *p* < 0.05 and ## *p* < 0.01 versus HepG2 cells treated with PA and vehicle.

**Figure 5 ijms-23-09768-f005:**
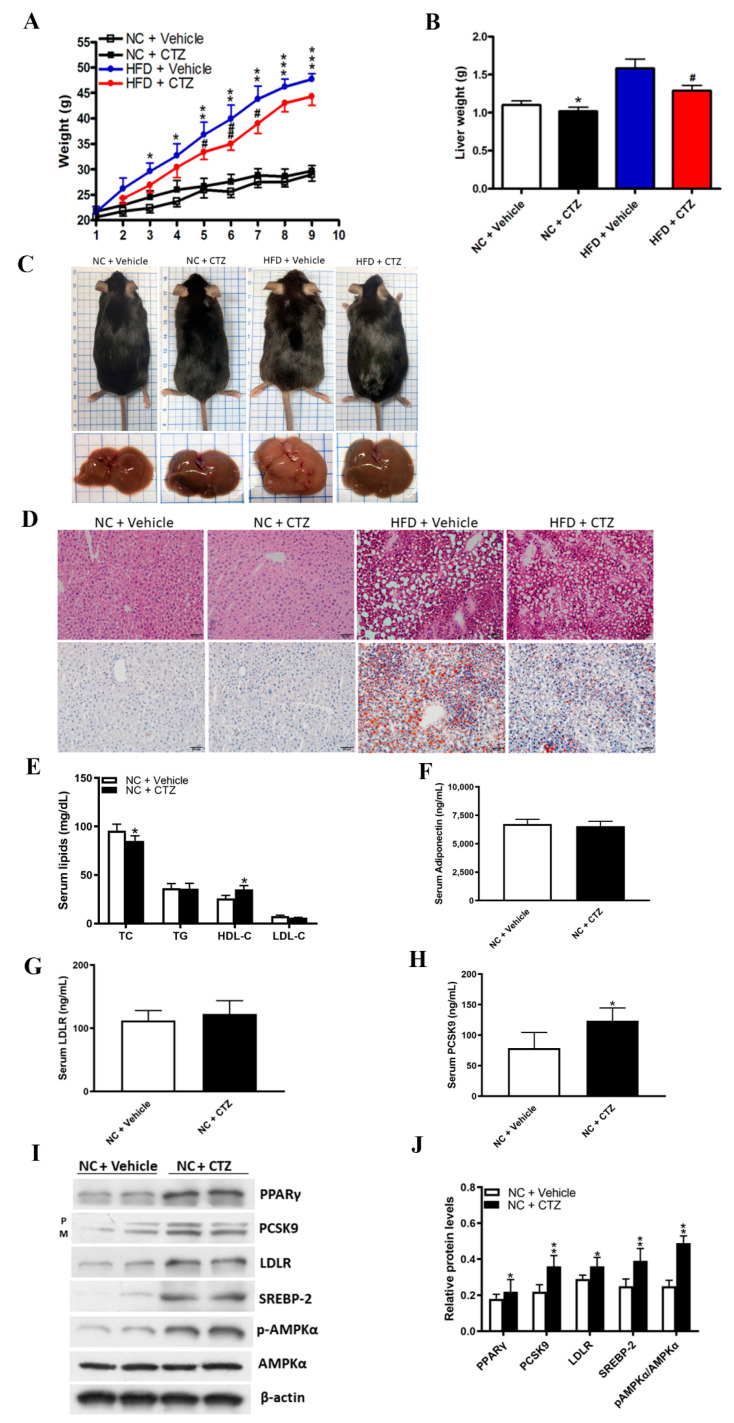
Effects of cilostazol on lipid homeostasis in the liver of mice fed with normal chow (NC) or high-fat diet (HFD) and lipid homeostasis-relevant profile in the serum or liver of the mice fed with NC. The mice were intraperitoneally injected with cilostazol (10 mg/kg of body weight) or vehicle twice a week for 12 weeks. (**A**,**B**) Quantification of body weight and liver weight (gm) in the NC-fed and HFD-fed mice treated with cilostazol or vehicle. * *p* < 0.05, ** *p* < 0.01 and *** *p* < 0.001 comparison between the vehicle-treated group fed with NC or HFD. # *p* < 0.05 and ## *p* < 0.01 comparison between the HFD-fed mice treated with cilostazol or vehicle. (**C**) Representative liver specimens obtained 24 weeks after treatment with cilostazol or vehicle. (**D**) Magnification 200×. The hematoxylin and eosin (upper) and Oil Red O staining (lower) of liver sections from NC-fed and HFD-fed mice treated with cilostazol or vehicle. Abundant accumulation of lipid droplets accompanied with hepatocyte damage in HFD-fed mice. This hazard effect could be diminished by cilostazol treatment. (**E**–**H**) Quantification of serum levels of lipid profile, adiponectin, PCSK9, and LDLR in NC-fed mice treated with cilostazol or vehicle. * *p* < 0.05 versus the NC-fed mice of vehicle-treated group. (**I**) Protein expression of PPARγ, PCSK9, LDLR, SREBP-2, and phosphorylated AMPKα were measured by Western blotting in the liver of the mice fed with NC. β-actin was used as a loading control. (**J**) Quantification of blot intensity relative to loading control is shown. * *p* < 0.05 and ** *p* < 0.01 versus the control group.

**Figure 6 ijms-23-09768-f006:**
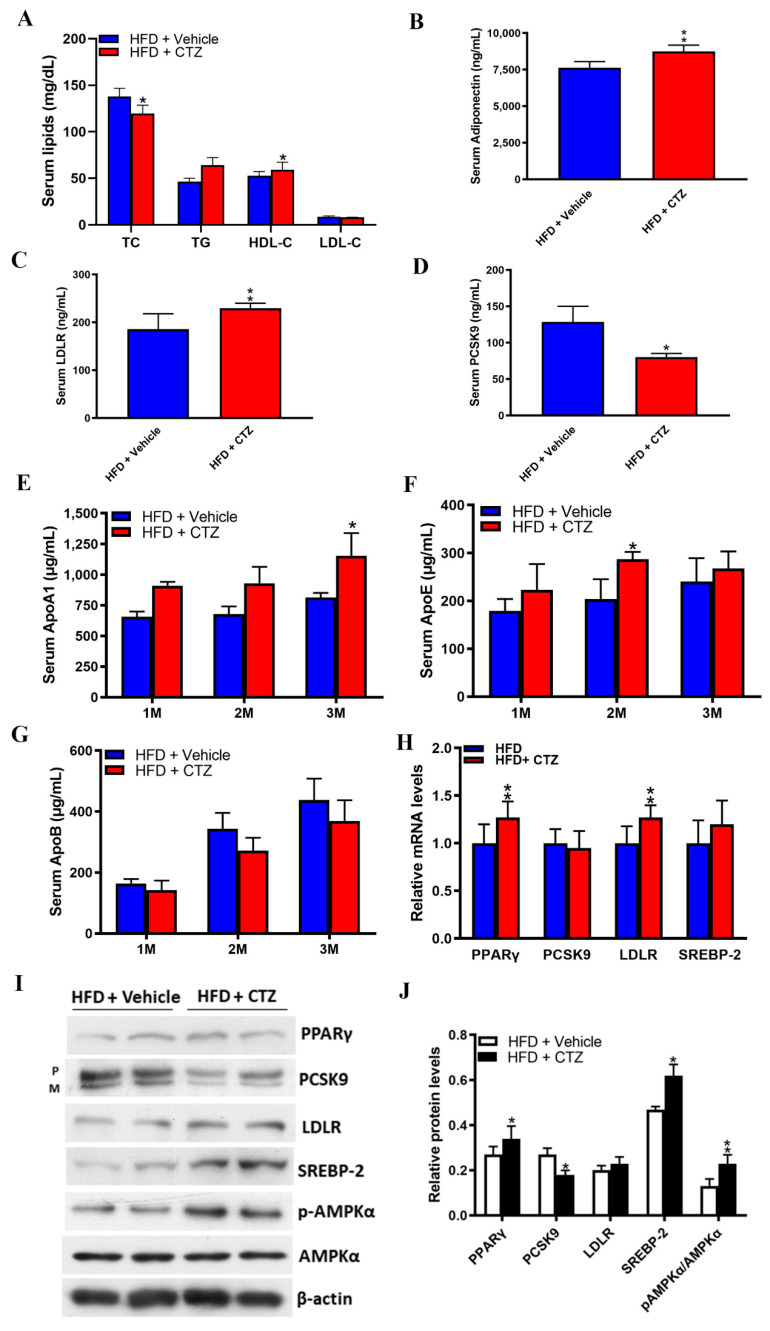
Effects of cilostazol on lipid homeostasis-relevant profile in the serum or liver of the mice fed with HFD. (**A**–**D**) Quantification of serum levels of lipid profile, adiponectin, PCSK9, and LDLR in HFD-fed mice treated with cilostazol or vehicle. * *p* < 0.05 versus vehicle-treated group. ** *p* < 0.01 versus vehicle-treated group. (**E**–**G**) Quantification of serum levels of ApoA1, ApoB, and ApoE in HFD-fed mice treated with cilostazol (red bar) or vehicle (blue bar). * *p* < 0.05 versus vehicle-treated group. ** *p* < 0.01 versus vehicle-treated group. Mouse serum Apo A1, Apo B, and Apo E levels were detected using ELISA kit. (**H**) Quantification of relative mRNA expression in the cilostazol-treated and vehicle-treated groups. ** *p* < 0.01 versus the vehicle-treated group. (**I**) Protein expression of PPARγ, PCSK9, LDLR, SREBP-2, and phosphorylated AMPKα were measured by Western blotting in the liver of the mice fed with HFD. β-actin was used as a loading control. (**J**) Quantification of blot intensity relative to loading control is shown. * *p* < 0.05 and ** *p* < 0.01 versus the vehicle-treated group.

**Figure 7 ijms-23-09768-f007:**
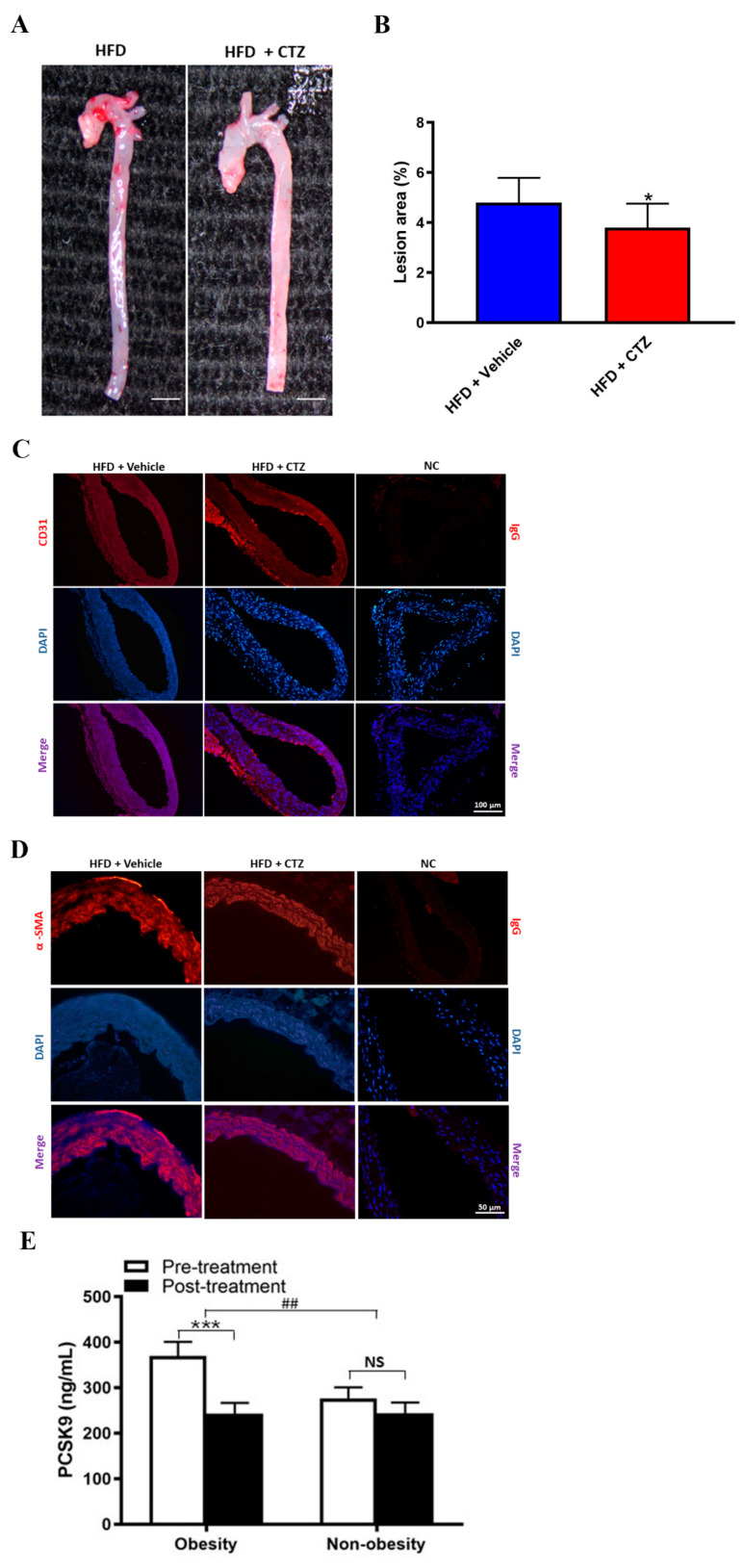
The anti-atherosclerotic effects of cilostazol in HFD-fed mice. (**A**,**B**) Representative images of aorta specimens from HFD-fed mice treated with vehicle (left) or cilostazol (right), and quantification of lesion area (%) in HFD-fed mice treated with cilostazol (red bar) and those without (blue bar). * *p* < 0.05 versus HFD-fed mice treated with vehicle. (**C**,**D**) Representative images of immunofluorescence staining with CD31, α-smooth muscle actin (α-SMA) (red), DAPI (nucleus, blue), and their colocalization (merge). Scale bar = 100 μM (**C**) and 50 μM (**D**). (**E**) Cilostazol treatment significantly reduced serum PCSK9 levels in patients with obesity (369.75 ± 30.94 vs. 243.27 ± 23.36 ng/mL, *p* < 0.001) but not in those without (276.26 ± 24.57 vs. 243.98 ± 32.14 ng/mL, *p* = 0.178). The changes in the serum concentrations of PCSK9 post-treatment between obese and non-obese patients were statistically significant (ΔPCSK9: 126.48 ± 24.93 vs. 32.28 ± 23.68 ng/mL, *p* = 0.009). *** *p* < 0.001 and NS: not significant (*p* > 0.05), comparison between groups. ## *p* < 0.01, comparison between obesity and non-obesity.

**Figure 8 ijms-23-09768-f008:**
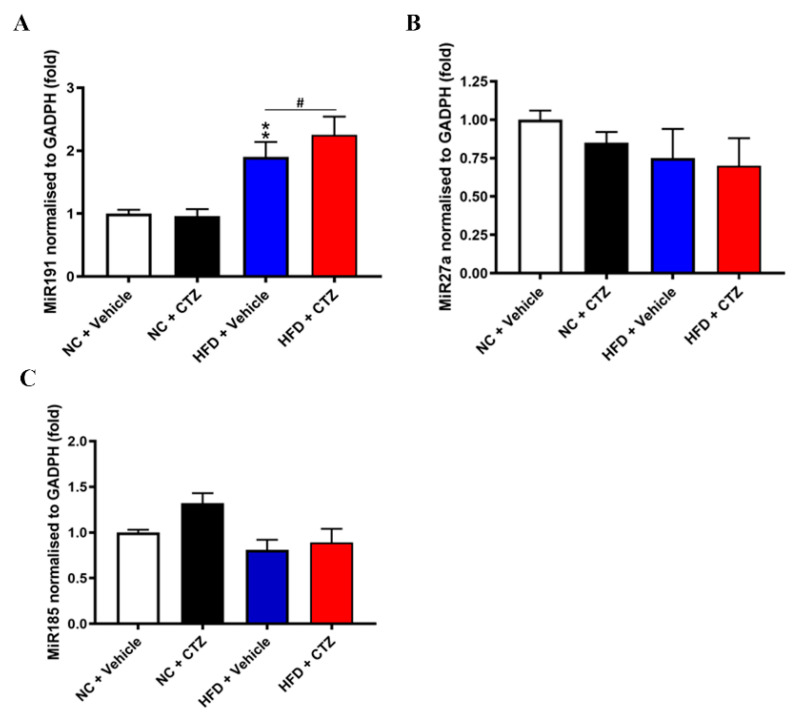
Effects of cilostazol on lipid metabolism-relevant miRNAs in the liver of the mice fed with NC and HFD. Levels of miRNAs were calculated from the results of real-time PCR and the volume of total RNA to compare the simplified absolute quantity in each component. (**A**–**C**) Quantification of miRNAs levels in the liver of the mice fed with NC and HFD treated with cilostazol or vehicle. ** *p* < 0.01 versus the NC-fed mice in vehicle-treated group. # *p* < 0.05, comparison between vehicle and cilostazol treatment in the HFD-fed mice.

## Data Availability

Not applicable.

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
