# Peer review of "Diverse Effects of Cilostazol on Proprotein Convertase Subtilisin/Kexin Type 9 between Obesity and Non-Obesity"

_ijms, 2022, doi:10.3390/ijms23179768_

Round 1
Reviewer 1 Report (Previous Reviewer 1)
The authors have added a number of new experiments to improve the overall merit of the study. I have no further concerns.
Author Response
Thanks for your favorable decision on our paper.
Reviewer 2 Report (New Reviewer)
The authors describe their work on the potential mechanisms of action of cilostazol on the expression of PCSK9 and lipid homeostasis. It was found that, in vitro, cilostazol treatment increased PCSK9 expression in vehicle-treated HepG2 cells, but decreased PCSK9 expression in palmitic acid-treated HepG2 cells. In vivo experiments revealed that cilostazol treatment increased serum levels of PCSK9 in normal mice but significantly reduced PCSK9 levels in obese mice. The expressions of PCSK9-relevant microRNAs also showed similar results. Clinical data showed that cilostazol treatment significantly reduced serum PCSK9 levels in patients with obesity. It was concluded that the obesity-dependent effects of cilostazol on PCSK9 expression observed from bench to bed-side demonstrates the therapeutic potential of cilostazol in clinical settings. This is an interesting study. Appropriate methodology has been employed and the manuscript is written very well. The authors are to be commended on the wealth of data generated. The conclusions appear to be justified based on the data at hand. I have some minor recommendations for consideration.
1. Title. Not sure about the subheading “obesity or not”. Can the title be modified?
2. Introduction. Can the authors provide a clear hypothesis to be tested?
3. Methods. Please provide more detail on the content/composition of the NC and HFD diets along with the caloric content. Was the daily intake of food the same across groups?
4. Results. Some of the figures (bar graphs and blot images) are a tad small and difficult to read. Can the authors please enlarge them?
5. Discussion. The authors should elaborate and emphasize the clinical applicability and relevance of their findings.
Author Response
Please see the attachment.

This manuscript is a resubmission of an earlier submission. The following is a list of the peer review reports and author responses from that submission.
Round 1
Reviewer 1 Report
The authors have previously reported that cilostazol can increase blood level of PCSK9 (Oncotarget. 2017 Nov 14;8(64):108042-108053). In this study, they analyzed the effects of cilostazol in HepG2 cells and HFD mice. However, the regulation of cilostazol on PCSK9 is completely opposite in HepG2 cells and HFD mice. Cilostazol can increase PCSK9 in HepG2 cells but decrease PCSK9 in HFD mice. This study fails to provide any mechanistic insight about why there is such a drastic difference. Without such critical information, the whole study does not have enough scientific merit to aid in understanding the action of cilostazol. In addition, several conclusions such as cilostazol regulating PPARgamma and improving atherosclerosis have been reported by many other groups. In this regard, a lot of data shown in this paper are not novel enough.
Reviewer 2 Report
Review of manuscript reference ijms-1738844; title: Diverse effects of cilostazol on proprotein convertase subtilisin/kexin type 9: obesity or not, by Chen et al.
General comment: the study reports the effects of phosphodiesterase 3 inhibitor cilostazol on PCSK9 and LDLR in a condition of obesity. Focusing in serum and liver, the authors investigate regulation of the expression of the protein in cultured cells, and correlate the results with those of adiponectin and other parameters, to finally confirm the potential beneficial effect in a mice experiment and a clinical assay with obese patients. The hypothesis is sound and objectives are clearly exposed and attractive; the study combines cell culture with live animals and humans making it a rather comprehensive work. The results are sometimes contradictory but the presentation is clear and they are fairly discussed. Some specific comments are detailed below:
Specific comments:
1) The authors utilize adenocarcinoma cell line HepG2 as a cell culture model to investigate the effects of cilostazol on hepatic LDLR and PCSK9, but the model is not properly introduced and there is no explanation for its selection considering that it is a transformed cell line.
2) Figure 1, incomplete statistical analysis; data from different doses should be compared among them, not only to control/untreated, in order to ensure a dose-dependent response. Until all comparisons have been performed, the authors should not state in lines 92 and 94 that there is a dose- or concentration-dependent response. The authors should be aware that a different number of asterisks (a different p) does not indicate a statistically significant difference between two data.
3) Figure 2, statistical analysis missing in panels A and B; although line 143 in text states that expression of LDLR protein was enhanced following cilostazol treatment in a dose-dependent manner, there is no indication of statistical analysis in the panels.
4) Figure 3, densitometry should be performed in protein bands panels D and E and average data should be exposed in bar diagrams including statistical comparisons.
5) Figure 5, incorrect statement; although line 230 in text says that cilostazol treatment was associated with lower serum ApoB levels, there is no statistical evidence in panel G of the significant decrease.
6) Lines 240-246; it seems rather unusual that a complete clinical assay was organized just to evaluate the serum levels of PCSK9. In fact, clinical assay is inserted in the middle of all data from mice experiments in last panel of figure 5 with no previous introduction, to continue with more animal data in figure 6. The authors should render a comprehensive explanation for the inclusion of a study with more than 100 patients to expose such a limited data.
7) Lines 345-347, summary before materials and methods; the authors conclude in these lines that: mechanisms of action responsible for cilostazol-related favorable effects on the lipid profile and anti-atherosclerosis in obese conditions by reducing PCSK9… despite the facts that CTZ increases PCSK9 in liver cells (figure 1), CTZ increases PCSK9 in serum of NC-fed mice (figure 4), and CTZ increases PCSK9 in liver of fat mice (figure 5I). The authors should explain this contradictory statement.
8) Line 437; protocol reference number for the animal assay should be included.
Round 2
Reviewer 1 Report
The authors did not perform any new experiment to improve the study.
Reviewer 2 Report
The authors have conveniently addressed all my comments and queries; thus, I recommend to accept the revised version for publication at IJMS.